# Individual Fairness in Feature-Based Pricing for Monopoly Markets

**Shantanu Das**[1]    **Swapnil Dhamal**[2]    **Ganesh Ghalme**[3]    **Shweta Jain**[4]    **Sujit Gujar**[1]

[1]Machine Learning Lab, International Institute of Information Technology, Hyderabad, India
[2]Télécom SudParis, Institut Polytechnique de Paris, Évry, France
[3]Indian Institute of Technology, Hyderabad, India
[4]Indian Institute of Technology, Ropar, India

## Abstract

We study fairness in the context of feature-based price discrimination in monopoly markets. We propose a new notion of individual fairness, namely, $\alpha$-fairness, which guarantees that individuals with similar features face similar prices. First, we study discrete valuation space and give an analytical solution for optimal fair feature-based pricing. We show that the cost of fair pricing is defined as the ratio of expected revenue in optimal feature-based pricing to the expected revenue in an optimal fair feature-based pricing (CoF) can be arbitrarily large in general. When the revenue function is continuous and concave with respect to the prices, we show that one can achieve CoF strictly less than 2, irrespective of the model parameters. Finally, we provide an algorithm to compute fair feature-based pricing strategy that achieves this CoF.

## 1 INTRODUCTION

The Internet has transformed the way markets function. Today's Internet-based ecosystems, such as entertainment and e-commerce marketplaces, are more consumer-centric and information-driven than ever. Data and AI systems are primarily used to power advertising, consumer retention, and personalized experience. These AI systems are deployed to aggregate individual choices and preferences to make personalized experiences possible. It is a common practice to use aggregated information about consumers to offer different prices to different consumers or segments of the market; this practice is commonly termed *price discrimination* [Varian, 1992].

Price discrimination has come under ethical scrutiny in multiple instances in the recent past. For example, it was found that Orbitz, an online travel agency, charges Mac users more than Windows users [Mattioli, 2012]. Uber's strategy to charge personalized prices came under heavy consumer backlash [Dholakia, 2015, Mahadawi, 2018], and thanks to the fine-grained data analysis of consumer behavior, several such instances were reported in the e-commerce and retail industry [Hinz et al., 2011]. More recently, Pandey and Caliskan [2021] showed that neighborhoods with high non-white populations, higher poverty, younger residents and high education levels faced higher cab trip fares in Chicago. Not surprisingly, the regulatory bodies and research community has taken notice. Economists have raised concerns on fairness issues of personalized pricing [Michel, 2016]. Price discrimination based on nationality or residence is illegal in the EU [2020]. In the USA, a white house report provides guidelines for enforcing existing anti-discrimination, privacy, and consumer protection laws while practicing discriminatory pricing [White House, 2015]. Given the overwhelming evidence and rising concerns, there is an urgent need to formally study price discrimination and fairness.

Sellers or firms use price discrimination for multiple reasons, including increasing revenue, covering transportation and storage costs, increasing market reach, rewarding loyal consumers, promoting a social cause, and so on [Cassady, 1946]. In general, price discrimination does not always raise ethical, and fairness issues and hence requires a careful inspection to categorize situations where this practice may lead to treatment disparity and invite regulatory intervention [Alan, 2020]. In this work, we focus on designing the pricing strategies for a seller (monopolist) who wants to maximize the revenue via price discrimination while ensuring fairness amongst the consumers.

A revenue-maximizing seller with complete knowledge of consumer valuations without fairness consideration would charge each consumer her valuation for the product. This pricing strategy, otherwise called *first-degree* price discrimination, may result in wild price fluctuations and is considered unfair in general [Moriarty, 2021]. Also, in practice, sellers do not have full access to individual consumer valuations but may have a distribution over valuations through *features*. In such *feature-based pricing* (FP), the seller seg-

*Accepted for the 38th Conference on Uncertainty in Artificial Intelligence* (UAI 2022).

regates the market into segments through the consumer features. The seller's problem then reduces to finding optimal pricing for each segment [Bergemann et al., 2015, Cummings et al., 2020]. Such FP is referred to as *third-degree price discrimination* in the literature. In this paper, our goal is to ensure fairness issues in feature-based personalized pricing.

Cohen et al. [2021] provide a notion of fairness among consumers' groups called *Price Fairness* under the assumption that the consumer valuations are known to the seller. In this paper, we consider the problem where consumers' valuations are unknown, and the seller uses the features to offer prices to the consumers.

**Our Contributions** We introduce the notion of $\alpha$-fairness in price discrimination, ensuring that similar individuals face similar prices. Following the definition of individual fairness, proposed by Dwork et al. [2012], we emphasize that if individuals with similar features are charged differently by segregating them into different segments, the interpersonal price comparison based on their features renders fairness issues. With this, we introduce a model for optimal *fair feature-based pricing* (FFP) as the problem of maximizing revenue while ensuring $\alpha$-fairness. We begin with two segments in the market and discrete valuations and propose an optimal FFP scheme (Section 4.2). To quantify the loss in the revenue due to fairness, we then introduce *cost of fairness* (CoF) – the ratio of expected revenue in an optimal FFP to the expected revenue in an optimal FP. We prove that a constant lower bound on CoF is generally impossible to achieve.

Next, in Section 5.1, under the assumption that the revenue function is concave in offered prices [Bergemann et al., 2021][1], we show that one can achieve a constant upper bound on CoF. Here, we show that the seller can compute optimal FFP using a convex program if it has access to distributional information (knows all consumers' valuation distribution functions). We then identify a class of FFP strategies, namely LINP-FFP that satisfies $\alpha$-fairness. With the help of these pricing strategies, we then show that the CoF is strictly less than 2 irrespective of model parameters. Finally, we propose OPT-LINP-FFP, an $O(K \log(K))$ time algorithm where $K$ is the number of segments that does not need access to complete distributional information and computes $\alpha$-fair pricing that achieves the aforementioned CoF (Algorithm 1 and Theorem 7).

## 2 RELATED WORK

The impact of discriminatory pricing on consumer and seller surplus was first considered by Bergemann et al. [2015]

when the consumer characteristics are known to the seller. The authors proposed a method to provide the optimal market segmentation. Das et al. [2022] considered the problem setting where the consumer valuations are unknown to the seller. The generalized problem was then considered by Cummings et al. [2020] which extended the work of Bergemann et al. [2015] to the case where only partial information about the consumer's valuation was known to the seller.

When the valuations of the consumers are not known, Elmachtoub et al. [2019, 2021] propose feature-based pricing and provides bounds on the value generated using idealized personalized pricing and Feature-based pricing over Uniform pricing. The value of feature-based pricing depends on the correlation between valuations and consumer features. Huang et al. [2019] consider the first-degree price discrimination over the social network where the centrality measures in social networks determine the features of the consumers. They provide bounds on the value of network-based personalized pricing in large random social networks with varying edge densities. Our work follows a similar approach because we derive personalized pricing from the features. However, naive feature-based pricing can be very unfair to the consumers, as we show in Proposition 2. Our focus is to design feature-based pricing that is fair at the same time.

Recently, many questions have been raised on the ethical side of price discrimination methods. Moriarty [2021] strongly criticizes online personalized pricing and suggests that personalized prices compete unfairly for the social surplus created by transactions. Gerlick and Liozu [2020] points out the need to design personalized pricing with ethical considerations, which can provide win-win outcomes for both organizations and consumers. Richards et al. [2016] discusses that discriminatory pricing leads to the perception of unfairness amongst the consumers, which undermines the stability of retail platforms. They discuss that when consumers are involved in forming the prices, this leads to improved fairness perception, thus leading to better retentivity. Levy and Barocas [2017] discusses that web-based platforms typically use many private features of user profiles to connect buyers and sellers. Users interacting on such platforms leads to discrimination regarding race, gender, and possibly other protected characteristics. All these studies lead to understanding the optimal price discriminatory strategies under the fairness constraint, which is the focus of our work.

Finally, Kallus and Zhou [2021] presents a list of metrics like *price disparity*, *equal access*, *allocative efficiency fairness* to measure and analyze fairness in feature-based pricing and study its interplay with welfare. The metrics discussed are mainly the group fairness notions which are entirely different from $\alpha$-fairness discussed in this paper. We emphasize that though the above papers discuss the ethical issues in price discrimination, none of them provides a

---

[1]This assumption is standard in economics as a large number of probability distributions follow this.

systematic approach to designing the pricing strategy that maximizes the revenue and ensures the fairness guarantee.

# 3 PRELIMINARIES

We consider a market with a monopolist seller seeking to price a single product available in infinite supply. The market is divided into finite number of segments $\mathcal{X} = \{x_1, x_2, \ldots, x_K\}$, where $x_i$ represents the $i^{\text{th}}$ segment. The seller, given access to $\mathcal{X}$, can choose to price discriminate across segments to extract maximum revenue.

Consumers' valuations for the single product are non-negative random variables drawn from the set $\mathcal{V}$ (same across all segments). Let $\mathcal{F}_i(\cdot)$ be the cumulative distribution function for the valuation of the consumers in $i^{\text{th}}$ segment, and $f_i(\cdot)$ be corresponding probability density function (probability mass function when $\mathcal{V}$ is discrete). In this paper, we consider the following two cases separately, (a) $\mathcal{V}$ is discrete and finite, and (b) $\mathcal{V}$ is continuous. Next, we present feature-based pricing model.

## 3.1 FEATURE-BASED PRICING MODEL

In feature-based pricing (FP), one can consider, without loss of generality, that the consumer feature is a representative of the market segment to which she belongs. Note that multiple consumers may have the same feature vector and all the consumers having identical features belong to the same market segment. For simplicity, we will write $p_i :=$ price offered to the consumer in the $i^{\text{th}}$ segment. A consumer makes the purchase only if her valuation is equal to or more than the offered price. The expected revenue per consumer generated from the $i^{\text{th}}$ segment with a price $p_i \in \mathbb{R}_+$ is given by

$$\pi_i(p_i) = p_i \cdot (1 - \mathcal{F}_i(p_i)) \tag{1}$$

Whenever it is clear from the context we refer to expected revenue per consumer from a segment to be expected revenue from that segment. Let $\beta_i$ be the fraction of consumers in the $i^{\text{th}}$ segment, then the expected revenue per consumer generated across all segments is given as $\Pi(\mathbf{p}) = \sum_{x_i \in \mathcal{X}} \beta_i \pi_i(p_i)$. We assume that $\beta_i$s are known to the seller. We call the sellers problem of revenue maximization as $\text{OPT}_{FP}(\mathcal{V}, \mathcal{X}, \mathcal{F}, \beta)$ where $\mathcal{F} = (\mathcal{F}_1, \ldots, \mathcal{F}_K)$ and $\beta = (\beta_1, \ldots, \beta_K)$.

In the absence of fairness constraints, $\text{OPT}_{FP}(\cdot)$ reduces to charging each segment separately and optimal FP strategy $\widehat{\mathbf{p}}$ consisting $\widehat{p}_i$ for segment $i$ is given by $\widehat{p}_i \in \underset{p_i \in \mathbb{R}_+}{\operatorname{argmax}} \pi_i(p_i)$.

**Fairness in Feature-based Pricing**   Let $d : \mathcal{X} \times \mathcal{X} \to \mathbb{R}_+$ be a distance function over $\mathcal{X}$. We assume that such a function exists and is well defined in $\mathcal{X}$, i.e., $(\mathcal{X}, d)$ is a metric space. The distance function quantifies the dissimilarity between feature vectors of individuals belonging to market segments. For simplicity we write $d(x_i, x_j) := d_{ij}$. Individual fairness in FP strategy is defined as:

**Definition 1** ($\alpha$-fairness).  A price function $\mathbf{p} : \mathcal{X} \to \mathbb{R}_+^K$ is $\alpha$-fair with respect to $d$ iff for all $x_i, x_j \in \mathcal{X}$, we have

$$|p_i - p_j| \leq \alpha \cdot d_{ij}. \tag{2}$$

We call a pricing strategy Fair Feature-based Pricing ($\alpha$-FFP) that satisfies Eq. (2) with a given value of $\alpha$. It is easy to see from the definition that any $\alpha$-FFP is also $\alpha'$-FFP for any $\alpha' \geq \alpha$. We will drop the quantifier $\alpha$ and call it FFP when it is clear from the context.

**Cost of Fairness (CoF)**   Next, we define CoF as the deviation from optimality due to fairness constraints given in Eq. (2). It is defined as the ratio of expected revenue generated by optimal feature-based pricing and fair feature-based pricing.

**Definition 2** (COST OF FAIRNESS (COF)).  Cost of fairness for an FFP strategy $\mathbf{p}$ is defined as

$$\text{CoF} = \frac{\Pi(\widehat{\mathbf{p}})}{\Pi(\mathbf{p})}. \tag{3}$$

In the following sections, we analyze FP and FFP strategies and their CoF when $\mathcal{V}$ is discrete (Section 4) and continuous (Section 5).

# 4 FFP FOR DISCRETE VALUATIONS

We want to ensure $\alpha$-fairness in the pricing strategy given the optimal FP. $\alpha$-fairness is achieved by maximizing revenue while satisfying the fairness constraints. In this section, we derive optimal FP (Section 4.1), propose how to achieve $\alpha$-fairness (Section 4.2), and provide an upper bound on CoF (Section 4.3) for discrete valuation setting.

We consider the simplest setting described as follows: Let the consumer segments be given by $\mathcal{X} = \{x_1, x_2\}$ and their valuations are drawn from a discrete set $\mathcal{V} = \{v_1, v_2\}$, we assume $v_1 < v_2$ without loss of generality. Let $\beta_1 = \beta$ and $\beta_2 = 1 - \beta$. Further, let $f_1(v_1) = q_1$ ($f_2(v_1) = q_2$) denote the probability that a consumer has valuation $v_1$ in segment 1 (segment 2). The expected revenue generated by $\mathbf{p}$ is given by:

$$\begin{aligned}\Pi(\mathbf{p}) =& \beta p_1 [q_1 \mathbb{1}(v_1 \geq p_1) + (1 - q_1) \mathbb{1}(v_2 \geq p_1)] \\ &+ (1 - \beta) p_2 [q_2 \mathbb{1}(v_1 \geq p_2) + (1 - q_2) \mathbb{1}(v_2 \geq p_2)]\end{aligned} \tag{4}$$

## 4.1 OPTIMAL FEATURE-BASED PRICING

As discussed earlier, $\Pi(\mathbf{p})$ can be maximized by maximizing $\pi_i(p_i)$ for each market segment independently if there

| Notation | Description |
|---|---|
| FP | Feature-based Pricing |
| FFP | Fair Feature-based Pricing |
| $\mathcal{F}_k, f_k()$ | Valuations CDF, PDF for $k^{\text{th}}$ consumer segment respectively |
| $\mathcal{X}$ | Set of all consumer features/types |
| $\mathcal{V}$ | Support set of consumers' valuations |
| $x_k$ | Consumer feature of the $k^{\text{th}}$ segment |
| $\beta_k$ | The fraction of consumers in the $k^{\text{th}}$ segment |
| $\mathbf{p} = (p_1, p_2, \ldots p_K)$ | Feature-based price vector |
| $\pi_k(p_k)$ | Revenue generated per consumer in the $k^{\text{th}}$ segment |
| $\Pi(p)$ | Revenue generated by $p$ across all consumer segments |
| $\widehat{\mathbf{p}} = (\widehat{p}_1, \widehat{p}_2, \ldots \widehat{p}_K)$ | Price function in optimal price discrimination |
| $d_{ij} := d(x_i, x_j)$ | A real-valued metric on the consumer feature space $\mathcal{X}$ |
| $\alpha$ | Fairness parameter |
| $\mathbf{p}^\star = (p_1^\star, p_2^\star, \ldots p_K^\star)$ | Optimal fair feature-based price function |
| $\widetilde{\mathbf{p}} = (\widetilde{p}_1, \widetilde{p}_2, \ldots, \widetilde{p}_K)$ | Price vector for OPT-LinP-FFP |
| CoF | Cost of Fairness |
| $L_m$ | Linear approximation of concave revenue curve with $m$ as parameter |

Table 1: Notation Table

are no fairness constraints. This problem is an integer program with price for each consumer type being a discrete variable. The revenue generated depends on $\beta_i$ and $f_i(\cdot)$ ($\beta$, $q_1, q_2$ in the current simplest case). The optimal FP is then given as

$$\text{For } i \in \{1, 2\}: \ \widehat{p}_i = \begin{cases} v_1 & \text{if } q_i \geq 1 - \frac{v_1}{v_2} \\ v_2 & \text{otherwise} \end{cases} \quad (5)$$

*Proof.* For a market segment $i$, $\pi_i(v_1) = v_1$ and $\pi_i(v_2) = v_2(1 - q_i)$. So, $\widehat{p}_i = v_1$ if

$$\pi_i(v_1) \geq \pi_i(v_2) \implies v_1 \geq v_2(1 - q_i) \implies q_i \geq 1 - \frac{v_1}{v_2}$$

otherwise, $\widehat{p}_i = v_2$.

$\square$

Next, we analyze the fairness aspects of the above pricing strategy.

## 4.2 OPTIMAL FAIR FEATURE-BASED PRICING

Let $(\mathcal{X}, d)$ be a metric space. We model the Optimal fair feature-based pricing (FFP) problem as integer program which maximizes $\Pi(\mathbf{p})$ with $\alpha$-fairness constraints described in Eq.(2). We denote this problem as $\text{OPT}_{FFP}(\mathcal{V}, \mathcal{X}, d, \mathcal{F}, \beta, \alpha)$ and the corresponding optimal FFP strategy is denoted as $\mathbf{p}^\star$. First we make an interesting and very useful claim for binary valuations.

**Lemma 1.** *When $\mathcal{V} = \{v_1, v_2\}$, and if $\widehat{\mathbf{p}}$ is not $\alpha$-fair, OPT $_{FFP}(\mathcal{V}, \mathcal{X}, d, \mathcal{F}, \beta, \alpha)$ reduces to $\text{OPT}_{FP}(\widetilde{\mathcal{V}}, \mathcal{X}, \mathcal{F}, \beta)$ where $\widetilde{\mathcal{V}}$ is either $\{v_1\}$, or $\{v_2\}$, or $\{v_1, v_1 + \alpha d_{12}\}$.*

*Proof.* Let $(p_1, p_2)$ be the tuple of offered prices. Note that if $v_2 - v_1 \leq \alpha d_{12}$ or $\widehat{p}_1 = \widehat{p}_2$, then the optimal $\mathbf{p}^\star = \widehat{\mathbf{p}}$ with support $\{v_1, v_2\}$ and $\widehat{\mathbf{p}}$ will be trivially fair. We consider a more interesting case when $v_2 - v_1 > \alpha d_{12}$ and $\widehat{p}_1 \neq \widehat{p}_2$. In this case, the only candidate support sets for optimal fair pricing strategy are: $\{v_1\}$, $\{v_2\}$, $\{v_1, v_1 + \alpha d_{12}\}$, $\{v_2 - \alpha d_{12}, v_2\}$. The optimal FFP does not take values from the set $\{v_2 - \alpha d_{12}, v_2\}$ as the consumers with valuation $v_1$ would not make any purchase. Hence, the expected revenue with support $\{v_2 - \alpha d_{12}, v_2\}$ will be less than or equal to the expected revenue with support $\{v_2\}$. $\square$

We now relax the constraint of binary valuation and analyze the optimal fair pricing scheme for $n$ valuations. The consumer segments are $\mathcal{X} = \{x_1, x_2\}$ with $\beta_1 = \beta$ and $\beta_2 = 1 - \beta$, the valuations are drawn from the set $\mathcal{V} = \{v_1, v_2, \ldots, v_n\}$, and $f_1(v_i) = q_{i,1}$ and $f_2(v_i) = q_{i,2}$. This is a simple extension of the pricing problem, $\text{OPT}_{FP}(\mathcal{V}, \mathcal{X}, \mathcal{F}, \beta)$ modelled as an integer program where the prices are drawn from the set $\mathcal{V}$. If $\widehat{\mathbf{p}}$ is not $\alpha$-fair then, the corresponding $\text{OPT}_{FFP}(\mathcal{V}, \mathcal{X}, d, \mathcal{F}, \beta, \alpha)$ can be solved by reducing it to $\text{OPT}_{FP}(\widetilde{\mathcal{V}}, \mathcal{X}, \mathcal{F}, \beta)$ with $\widetilde{\mathcal{V}}$ given by:

$$\widetilde{\mathcal{V}} = \begin{cases} \{v_i\}, v_i \in \mathcal{V} & \text{if } p_1^\star = p_2^\star \\ \{v_j, v_j + \alpha d_{12}, v_j - \alpha d_{12}\}, v_j \in \mathcal{V} & \text{if } p_1^\star \neq p_2^\star \end{cases}$$

Given the set $\widehat{\mathcal{V}}$, the pricing problem $\text{OPT}_{FP}(\widetilde{\mathcal{V}}, \mathcal{X}, \mathcal{F}, \beta)$ can be solved in constant time. It is easy to see that computing $\widehat{\mathcal{V}}$ takes $\mathcal{O}(n^2)$ time for $n$ valuations and 2 consumer types. Therefore, the fair pricing problem $\text{OPT}_{FFP}(\mathcal{V}, \mathcal{X}, d, \mathcal{F}, \beta, \alpha)$ can be solved in $\mathcal{O}(n^2)$ time.

## 4.3 COF ANALYSIS

For $n = 2$, based on the values of $q_1, q_2$ we have the following cases:

1. $p_1^\star = p_2^\star = v_1$     3. $p_1^\star = v_1 + \alpha d_{12}, p_2^\star = v_1$
2. $p_1^\star = p_2^\star = v_2$     4. $p_1^\star = v_1, p_2^\star = v_1 + \alpha d_{12}$

In cases 1 and 2, optimal fair pricing is equivalent to uniform pricing and therefore are 'trivially' fair with CoF = 1, i.e., $\Pi(\widehat{\mathbf{p}}) = \Pi(\mathbf{p}^\star)$. For case 3, $\Pi(\widehat{\mathbf{p}})$ and $\Pi(\mathbf{p}^\star)$ are given as:

$$\Pi(\widehat{\mathbf{p}}) = \beta(v_2)(1 - q_1) + (1 - \beta)v_1$$
$$\Pi(\mathbf{p}^\star) = \beta(v_1 + \alpha d_{12})(1 - q_1) + (1 - \beta)v_1$$

Then the cost of fairness for case 3 is given as:

$$\begin{aligned} \text{CoF} &= \frac{\Pi(\widehat{\mathbf{p}})}{\Pi(\mathbf{p}^\star)} = \frac{\beta(v_2)(1 - q_1) + (1 - \beta)v_1}{\beta(v_1 + \alpha d_{12})(1 - q_1) + (1 - \beta)v_1} \\ &= \frac{\beta(v_2 - v_1) + v_1 - \beta v_2 q_1}{\beta \alpha d_{12}(1 - q_1) - \beta v_1 q_1 + v_1} \\ &= \frac{\beta\left(1 - \frac{v_1}{v_2}\right) + \frac{v_1}{v_2} - \beta q_1}{\beta\left(\frac{\alpha d_{12}}{v_2}\right)(1 - q_1) - \beta\left(\frac{v_1}{v_2}\right)q_1 + \frac{v_1}{v_2}} \end{aligned} \quad (6)$$

Replacing $\beta$ with $(1 - \beta)$ and $q_1$ with $q_2$ in the above expression, we get a similar approximation of CoF for case 4.

**Proposition 2.** *Cost of fairness with discrete valuations can go arbitrarily bad.*

*Proof.* From Eq. (6) when $\frac{v_1}{v_2} \to 0$, we have $\text{CoF} = \frac{v_2}{\alpha d_{12}}$. The CoF (in Case 3 and/or Case 4) is arbitrarily bad if $d_{12} > 0$ when there is a large difference between $v_1$ and $v_2$. Note that $d_{12} = 0$ is uninteresting as the seller is unable to distinguish between two segments. □

Note that $v_2$ being arbitrarily large need not be a typical setting. Hence, we work with bounded support valuations in the backdrop of the above negative results. In the next section, we make assumptions based on standard economic literature about the revenue functions $\pi_i(\cdot)$, i.e., concave revenue

functions and common support [Bergemann et al., 2021]. As argued in Section 3 of Dhangwatnotai et al. [2015], valuation distributions satisfying Monotone Hazard Rate (MHR) satisfy the assumptions mentioned above regarding revenue functions. It is also observed that the revenue functions are concave for another commonly analyzed family of distributions in literature called the regular distributions, in which the virtual valuation is non-decreasing (Section 4.3 of Bergemann et al. [2021]). MHR is a common assumption in Econ-CS [Hartline and Roughgarden, 2009]. Therefore, in the following section, we analyze the cost of fairness for such valuation distributions and the associated concave revenue functions.

## 5 FFP FOR CONTINUOUS VALUATIONS

In this section, we consider feature-based pricing with continuous valuations. We impose a standard restriction on the revenue functions $\pi_i(\cdot)$ such that they are concave on the common support $\mathcal{V} = [\underline{v}, \bar{v}]$ [Bergemann et al., 2021]. The consumer segments are identified by the associated feature vectors $x_i \in \mathcal{X}$. $\underline{v}$ is the marginal cost defined as a minimum feasible valuation for which a seller is willing to sell the product. The marginal cost may include the cost of production, transportation, etc. On the other hand, $\bar{v}$ is the maximum consumer valuation. Without loss of generality, we consider that maximum consumer valuation is greater than marginal cost; i.e., trade occurs.

We begin with a tight upper bound on the CoF under conditions as mentioned above (Section 5.1) followed by two pricing schemes based on the available information about the revenue functions (Section 5.2), and finally, we present an algorithm that achieves the CoF bound in Section 5.3.

### 5.1 OPTIMAL FFP FOR CONTINUOUS VALUATIONS

The problem of determining optimal FFP can be modeled as a convex program with $\alpha$-fairness as linear constraints. The convex program below describes $\text{OPT}_{FFP}(\mathcal{V}, \mathcal{X}, d, \mathcal{F}, \beta, \alpha)$ model with complete knowledge of revenue functions $\pi_i(\cdot)$.

$$\max_{p_k \in \mathcal{V}, \forall k} \Pi(\mathbf{p}) = \sum_{k=1}^{K} \beta_k \pi_k(p_k)$$
$$\text{subject to, } |p_i - p_j| \leq \alpha d(x_i, x_j), \forall i \neq j$$
$$p_i \geq 0, \forall i \in [K]$$

Let $\mathbf{p}^\star$ be a solution to the above problem.

## 5.2 LINP-FFP AND COF ANALYSIS

Let $D_i := \min_{j \neq i} d_{ij}$. With the following proposition, we propose a class of $\alpha$-fair pricing strategies.

**Proposition 3.** *For a given $m \in [\underline{v}, \overline{v}]$, if the price function satisfies $|p_i - m| \leq \frac{\alpha}{2} D_i$ for all $i \in [K]$ then it satisfies $\alpha$-fairness.*

*Proof.* From triangle inequality, we have $|p_i - p_j| \leq |p_i - m| + |p_j - m| \leq \frac{\alpha}{2} D_i + \frac{\alpha}{2} D_j \leq \alpha d_{ij}$. The last inequality results from the fact that $D_i = \min_{k \neq i} d_{ik} \leq d_{ij}$ and $D_j = \min_{k \neq j} d_{ik} \leq d_{ji} = d_{ij}$. $\square$

In other words, to ensure that the prices for different segments are not too different, it is enough to ensure that the pricing for each segment is not too different from some common point $m$. The pricing for all the segments would hence be around this point and could be determined with respect to this point. We term this point as *pivot*. We now present the second FFP model, an $\alpha$-fair pricing strategy that is pivot-based and satisfies the condition in Proposition 3, with access to only $\widehat{p}_i$ for a given $m$.

$$p_i = \begin{cases} m + \alpha D_i/2 & \text{if } \widehat{p}_i - m \geq \alpha D_i/2 \\ m - \alpha D_i/2 & \text{if } m - \widehat{p}_i \geq \alpha D_i/2 \\ \widehat{p}_i & \text{otherwise} \end{cases} \quad (8)$$

We call this pricing scheme LINP-FFP. It is easy to see that the above pricing strategy is $\alpha$-fair. We now present the COF bound for LINP-FFP.

**Theorem 4.** *The Cost of Fairness for optimal fair price discrimination with concave revenue functions satisfies*

$$\mathrm{CoF} \leq \frac{2}{1 + \min\left\{\alpha \frac{\min_i D_i}{\overline{v} - \underline{v}}, 1\right\}}$$

*Proof.* We prove that the above COF is satisfied by LINP-FFP and hence the theorem. Let $m \in [\underline{v}, \overline{v}]$ be a pivot point (See Figure 1). Let

$$\gamma_i := \begin{cases} \frac{(m - \underline{v}) + \alpha D_i/2}{\widehat{p}_i - \underline{v}} & \text{if } \widehat{p}_i - m \geq \alpha D_i/2 \\ \frac{(\overline{v} - m) + \alpha D_i/2}{\overline{v} - \widehat{p}_i} & \text{if } m - \widehat{p}_i \geq \alpha D_i/2 \\ 1 & \text{otherwise} \end{cases} \quad (9)$$

Let $\widehat{\pi}_i$ be the expected revenue generated from the $i^{\text{th}}$ segment under $\widehat{\mathbf{p}}$. We now show the following supporting lemma.

**Lemma 5.** *The pricing strategy given in Eq. (8) guarantees at-least $\gamma_i$ fraction of optimal revenue from segment $i$, i.e., $\pi_i \geq \gamma_i \widehat{\pi}_i$.*

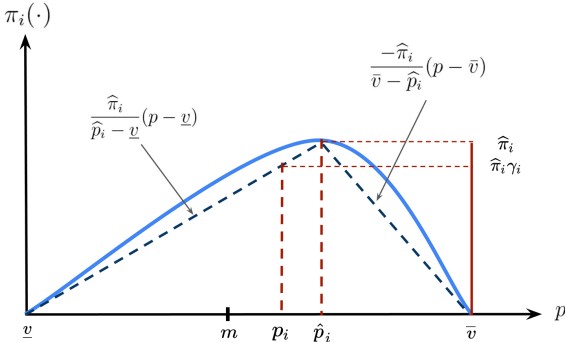

Figure 1: Concave revenue function $\pi_i(\cdot)$ and its linear approximation $L_i(\cdot)$ (arrows show equations for $L_i(\cdot)$). Figure represents the case $\widehat{p}_i - m \geq \alpha D_i/2$ for which LINP-FFP assigns $p_i = m + \alpha D_i/2$. The case $m - \widehat{p}_i \geq \alpha D_i/2$ is similar.

*Proof.* A lower bound to the concave revenue functions $\pi_i(\cdot)$ for any segment $i$ is the piecewise linear approximation $L_i$, given by (see Figure 1):

$$L_i(p) = \begin{cases} \frac{\widehat{\pi}_i}{\widehat{p}_i - \underline{v}}(p - \underline{v}), & p \leq \widehat{p}_i \\ \frac{-\widehat{\pi}_i}{\overline{v} - \widehat{p}_i}(p - \overline{v}), & p > \widehat{p}_i \end{cases} \quad (10)$$

So, for each consumer segment $i$ we have,

$$L_i(p) \leq \pi_i(p), \ \forall p \in [\underline{v}, \overline{v}]$$

Expected revenues generated per consumer in segment $i$ by pricing rule in Eq. (8) for $\widehat{p}_i - m \geq \alpha D_i/2$, $m - \widehat{p}_i \geq \alpha D_i/2$, and remaining cases are given below in the respective order

$$\pi_i(p_i) \geq L_i(p_i) = \frac{\widehat{\pi}_i}{\widehat{p}_i - \underline{v}}(m + \alpha D_i/2 - \underline{v}) = \widehat{\pi}_i \gamma_i$$

$$\pi_i(p_i) \geq L_i(p_i) = \frac{-\widehat{\pi}_i}{\overline{v} - \widehat{p}_i}(m - \alpha D_i/2 - \overline{v}) = \widehat{\pi}_i \gamma_i$$

$$\pi_i(p_i) = L_i(\widehat{p}_i) = \widehat{\pi}_i$$

This proves the lemma. $\square$

Let $\pi_i^\star$ denote the expected revenue generated from the $i^{\text{th}}$ segment by $\mathbf{p}^\star$. So, COF for optimal FPP is given by:

$$\mathrm{CoF} = \frac{\sum_{i \in [K]} \beta_i \widehat{\pi}_i}{\sum_{i \in [K]} \beta_i \pi_i^\star} \leq \frac{\sum_{i \in [K]} \beta_i \widehat{\pi}_i}{\sum_{i \in [K]} \beta_i \pi_i} \quad \text{(Optimality of } \pi_i^\star\text{)}$$

$$\leq \frac{\sum_{i \in [K]} \beta_i \widehat{\pi}_i}{\sum_{i \in [K]} \beta_i \gamma_i \widehat{\pi}_i} \quad \text{(Lemma 5)}$$

In order to prove the said COF bound, it suffices to show that there exists an $m$ (and hence a corresponding pricing

strategy using Eq. (8)) for which the said CoF bound is satisfied. It can be seen that for $m = (\underline{v} + \bar{v})/2$, and replacing denominators in Eq. (9) by $\bar{v} - \underline{v}$, we have that

$$
\begin{aligned}
\text{CoF} &\leq \frac{\sum_{i \in [K]} \beta_i \widehat{\pi}_i}{\sum_{i \in [K]} \beta_i \widehat{\pi}_i \left( \frac{1}{2} + \min\{ \frac{\alpha D_i}{2(\bar{v} - \underline{v})}, 1 \} \right)} \\
&\leq \frac{\sum_{i \in [K]} \beta_i \widehat{\pi}_i}{\left( \sum_{i \in [K]} \beta_i \widehat{\pi}_i \right) \left( \frac{1}{2} + \min\{ \frac{\alpha \min_j D_j}{2(\bar{v} - \underline{v})}, 1 \} \right)} \\
&= \frac{2}{1 + \min\left\{ \alpha \frac{\min_j D_j}{\bar{v} - \underline{v}}, 1 \right\}}
\end{aligned}
$$

$\square$

It is worth noting here that the cost of fairness does not depend on the number of the segments and the population distribution among these segments. So, if the segments are well separated in terms of the distance between features of consumers across segments, the number of segments, and the distribution of consumer population in these segments do not affect revenue guarantee. Also, if the admissible prices are supported over a large interval, the fairness guarantee becomes weaker. This insight discourages pricing schemes with wildly varying prices across segments. Finally, if $\alpha = 0$, i.e., without any fairness constraints, we recover the bound of 2 proved in Bergemann et al. [2021].

We emphasize that the bound is strictly less than 2 because, under fairness constraints, $\alpha \neq 0$ and typically, the consumer types are well separated in the feature space according to the metric $d$ else, the consumer types are indistinguishable for the seller hence, $d_{ij} \neq 0$ for all $i, j \in [K]$. This is an improvement of the CoF bound given in Bergemann et al. [2021].

**Tightness of CoF bound:** We claim that the CoF bound presented above is tight. In the following example, equality holds and proves the tightness of the bound.

**Example 1** (Tightness of the CoF bound). *Consider $K = 2$ where $\beta_1 = \beta_2 = \frac{1}{2}$. Consider $\mathcal{F}_i$ be such that $\pi_i(\cdot) = L_i(\cdot)$ with $\widehat{p}_1 = \underline{v} + \epsilon, \widehat{p}_2 = \bar{v} - \epsilon$, where $\epsilon \to 0$, and $\widehat{\pi}_1 = \widehat{\pi}_2$. It can be seen that if $\alpha$ is such that $\alpha d_{12} < \bar{v} - \underline{v}$, any FP satisfying $p_2 - p_1 = \alpha d_{12}$ and $p_1, p_2 \in [\widehat{p}_1, \widehat{p}_2]$ is an optimal FFP (fair FP), and the corresponding $\text{CoF} = \frac{2}{1 + \frac{\alpha d_{12}}{\bar{v} - \underline{v}}}$. If $\alpha d_{12} \geq \bar{v} - \underline{v}$, the optimal FP is $\alpha$-fair and so, $\text{CoF} = 1$. Hence, for this example, $\text{CoF} = \frac{2}{1 + \min\{\alpha \frac{d_{12}}{\bar{v} - \underline{v}}, 1\}}$. This shows the tightness of the CoF bound derived in Theorem 4.*

We now present an algorithm, OPT-LINP-FFP, to find the optimal pivot $m^\star$ in the above LINP-FFP strategy when only $\widehat{p}$ and $\widehat{\pi}_i$s are known.

## 5.3 PROPOSED ALGORITHM

As LINP-FFP satisfies $\alpha$-fairness (Proposition 3), and also achieves CoF bounds in Theorem 4, we look for a pricing strategy optimal within class of LINP-FFP. It reduces to finding an optimal pivot that maximizes revenue. In this section, we propose a binary-search-based algorithm for the same. For pricing $\mathbf{p}$, the expected revenue generated per consumer is given by $\Pi(\mathbf{p}) = \sum_{i=1}^{K} \beta_i \pi_i(p_i)$. Let $\tau_i := \frac{\alpha}{2} D_i$. Observe from Lemma 5 that $\Pi(\mathbf{p})$ is lower bounded as:

$$
\begin{aligned}
\Pi(\mathbf{p}) \geq \Pi_m(\mathbf{L}) = \sum_{i=1}^{K} \beta_i \gamma_i \widehat{\pi}_i &= \sum_{i:|\widehat{p}_i - m| < \tau_i} \beta_i \widehat{\pi}_i + \\
\sum_{i:\widehat{p}_i - m \geq \tau_i} \beta_i \widehat{\pi}_i \frac{m + \tau_i - \underline{v}}{\widehat{p}_i - \underline{v}} &+ \sum_{i:m - \widehat{p}_i \geq \tau_i} \beta_i \widehat{\pi}_i \frac{\bar{v} - m + \tau_i}{\bar{v} - \widehat{p}_i}
\end{aligned}
$$
(13)

### Determining Optimal Pivot $m$

As we can see, the revenue generated by LINP-FFP is lower bounded by a piecewise linear function in $m$. With the aim of achieving a better lower bound, we now address the problem of determining an optimal pivot $m^\star \in \underset{m \in [\underline{v}, \bar{v}]}{\arg\max} \Pi_m(\mathbf{L})$.

### Pricing Algorithm

In what follows, we call the candidate points $m$ for optimal pivot, i.e., for maximizing $\Pi_m(\mathbf{L})$, as *critical points*. We denote the set of these critical points as $\mathcal{M}$.

**Lemma 6.** $\Pi_m(\mathbf{L})$ *as a function of $m$ is concave and piecewise linear with the set of critical points $\mathcal{M} = \left( \{\widehat{p}_i - \frac{\alpha}{2} D_i, \widehat{p}_i + \frac{\alpha}{2} D_i\}_{i \in [K]} \cap [\underline{v}, \bar{v}] \right) \cup \{\underline{v}, \bar{v}\}$.*

*Proof.* It is easy to see that for a segment $i$, $\gamma_i$ as a function of $m$ is continuous and piecewise linear with breakpoints (i.e., points at which piecewise linear function changes slope): $\widehat{p}_i - \frac{\alpha}{2} D_i$ and $\widehat{p}_i + \frac{\alpha}{2} D_i$ provided they are in the range $[\underline{v}, \bar{v}]$. The set of breakpoints is hence $\{\widehat{p}_i - \frac{\alpha}{2} D_i, \widehat{p}_i + \frac{\alpha}{2} D_i\} \cap [\underline{v}, \bar{v}]$. Also, the slope monotonically decreases at the breakpoints, i.e., $\gamma_i$ is a concave function of $m$.

From Eq. (13), we can see that $\Pi_m(\mathbf{L})$ is a weighted sum over all segments, of $\gamma_i$'s with constant weights $\beta_i \widehat{\pi}_i$. So, $\Pi_m(\mathbf{L})$ as a function of $m$ is concave and piecewise linear with breakpoints belonging to the following set: $\{\widehat{p}_i - \frac{\alpha}{2} D_i, \widehat{p}_i + \frac{\alpha}{2} D_i\}_{i \in [K]} \cap [\underline{v}, \bar{v}]$. Hence, a point $m$ that maximizes $\Pi_m(\mathbf{L})$ belongs to either the aforementioned set of breakpoints, or the set of its boundary points $\{\underline{v}, \bar{v}\}$. Thus, the set of critical points $\mathcal{M} = \left( \{\widehat{p}_i - \frac{\alpha}{2} D_i, \widehat{p}_i + \frac{\alpha}{2} D_i\}_{i \in [K]} \cap [\underline{v}, \bar{v}] \right) \cup \{\underline{v}, \bar{v}\}$. $\square$

Our algorithm OPT-LINP-FFP (Optimal Linearized Pivot-based Fair Feature-based Pricing) which determines an optimal pivot $m^\star$ and provides an $\alpha$-fair pricing strategy ($\widetilde{\mathbf{p}}$) is presented in Algorithm 1.

---

**Algorithm 1:** OPT-LINP-FFP

**Input:** $\alpha, \{(\widehat{p}_i, \widehat{\pi}_i, \beta_i, D_i)\}_{i=1}^K$

**Output:** $m^\star, \widetilde{\mathbf{p}}$

   /* Creating and sorting the set of
      critical points                    */

1   $\mathcal{M} \leftarrow \{\underline{v}, \bar{v}\}$

2   **for** $i \in [K]$ **do**

3      $\tau_i \leftarrow \frac{\alpha}{2} D_i$

4      **if** $\widehat{p}_i - \tau_i > \underline{v}$ **then**

5         $\mathcal{M} \leftarrow \mathcal{M} \cup \{\widehat{p}_i - \tau_i\}$

6      **if** $\widehat{p}_i + \tau_i < \bar{v}$ **then**

7         $\mathcal{M} \leftarrow \mathcal{M} \cup \{\widehat{p}_i + \tau_i\}$

8   sort($\mathcal{M}$)

   /* Binary search for optimal pivot      */

9   $\ell \leftarrow 0, r \leftarrow |\mathcal{M}| - 1$

10   **while** $\ell \leq r$ **do**

11      $z \leftarrow \lfloor \frac{\ell+r}{2} \rfloor$     // $\mathcal{M}[z]$ is the current pivot

       /* Computing the expression in
           Eq. (13) at current and adjacent
           critical points            */

12      $\Pi_{\mathcal{M}[z-1]} \leftarrow 0, \Pi_{\mathcal{M}[z]} \leftarrow 0, \Pi_{\mathcal{M}[z+1]} \leftarrow 0$

13      **for** $y \leftarrow \{z-1, z, z+1\}$ **do**

14         **for** $i \leftarrow 1$ *to* $K$ **do**

15            **if** $\widehat{p}_i \geq \mathcal{M}[y] + \tau_i$ **then**

16               $\gamma_i \leftarrow \frac{\mathcal{M}[y] - \underline{v} + \tau_i}{\widehat{p}_i - \underline{v}}$

17           **else if** $\widehat{p}_i \leq \mathcal{M}[y] - \tau_i$ **then**

18               $\gamma_i \leftarrow \frac{\bar{v} - \mathcal{M}[y] + \tau_i}{\bar{v} - \widehat{p}_i}$

19           **else**

20               $\gamma_i \leftarrow 1$

21           $\Pi_{\mathcal{M}[y]} \leftarrow \Pi_{\mathcal{M}[y]} + \beta_i \gamma_i \widehat{\pi}_i$

22      **if** $\Pi_{\mathcal{M}[z-1]} \leq \Pi_{\mathcal{M}[z]} \leq \Pi_{\mathcal{M}[z+1]}$ **then**

23         $\ell \leftarrow z + 1$

24      **else if** $\Pi_{\mathcal{M}[z-1]} \geq \Pi_{\mathcal{M}[z]} \geq \Pi_{\mathcal{M}[z+1]}$ **then**

25         $r \leftarrow z - 1$

26      **else**

27         $m^\star \leftarrow \mathcal{M}[z]$

28         **break**

   /* Pricing for the different segments */

29   **for** $i \in [K]$ **do**

30      **if** $\widehat{p}_i \geq m^\star + \tau_i$ **then**

31         $\widetilde{p}_i \leftarrow m^\star + \tau_i$

32      **else if** $\widehat{p}_i \leq m^\star - \tau_i$ **then**

33         $\widetilde{p}_i \leftarrow m^\star - \tau_i$

34      **else**

35         $\widetilde{p}_i \leftarrow \widehat{p}_i$

---

**Theorem 7.** *The* OPT-LINP-FFP *algorithm (a) returns optimal pivot point* $m^\star$ *and runs in* $\mathcal{O}(K \log(K))$ *time, and (b) achieves the* CoF *bound given in Theorem 4.*

*Proof.* (a) The first module is the creation and sorting of the set of critical points $\mathcal{M}$, which takes $\mathcal{O}(K \log(K))$ time. Owing to Lemma 6, we can find an optimal pivot $m^\star$ using binary search over $\mathcal{M}$. Here, the number of critical points are at most $2K + 2$, i.e., $|\mathcal{M}| \leq 2K + 2$. So, in the second module that finds an optimal pivot, the binary search in the outer (*while*) loop runs for $\mathcal{O}(\log(|\mathcal{M}|))$ iterations, and the inner (*for*) loops run for $\mathcal{O}(K)$ iterations overall. Thus, the running time of the second module is $\mathcal{O}(K \log(K))$. The third module that computes pricing for the different segments runs in $\mathcal{O}(K)$ time. So, the total running time of Algorithm 1 is $\mathcal{O}(K \log(K))$.

(b) From Theorem 4, for $m = (\underline{v} + \bar{v})/2$, the CoF bound holds. Also, $\Pi_{m^\star}(\mathbf{L}) \geq \Pi_m(\mathbf{L})$ for all $m \neq m^\star$. We have:

$$\text{CoF} = \frac{\Pi(\widehat{\mathbf{p}})}{\Pi(\widetilde{\mathbf{p}})} \leq \frac{\Pi(\widehat{\mathbf{p}})}{\Pi_{m^\star}(\mathbf{L})} \leq \frac{\Pi(\widehat{\mathbf{p}})}{\Pi_m(\mathbf{L})}$$

This completes the proof of the theorem. $\qquad\square$

**Experiments on Synthetic Data:** We validate our theoretical claims on synthetically generated consumer valuations. The data is generated by approximating valuation distributions $f_k(\cdot)$ as triangular functions over a chosen common support by generating random peaks for each consumer segment $k$. We then find the revenue peaks $\widehat{\pi}_k$ and the corresponding $\widehat{p}_k$ values for OPT-LINP-FFP. The consumer features are $m$-dimensional random vectors where each entry is in the range $[0, 1]$ and the distance metric used is Euclidean 2-norm. We assume that $\beta_k = 1/n$ for all consumer types $k$ where, $n$ is the number of consumer types. On simulated data, OPT-LINP-FFP achieves CoF= 1.0806 (worst case CoF = 1.1834, average case CoF = 1.0806) with coefficient of variation = 0.027 for 500 iterations.

## 6 DISCUSSION

This paper built a foundation for the design of fair feature-based pricing by proposing a new fairness notion called $\alpha$-fairness. Our impossibility result on the discrete valuation setting restricted us from attaining a finite cost of fairness (CoF) in general settings. Interestingly, in the continuous valuation setting with concave revenue functions, we showed that a family of pricing schemes, LINP-FFP, provided a CoF strictly less than 2. Finally, we proposed an algorithm, OPT-LINP-FFP, which gave us an optimal pricing strategy within this family. Peaks of revenue distributions are sufficient statistics for computing optimal fair feature-based pricing. Compared with the regression models, which could also be used to learn optimal pricing, our approach requires significantly less information about the

distribution function. More specifically, we observe that peaks of revenue distributions are sufficient statistics for computing optimal fair feature-based pricing.

We leave the problem of finding an optimal segmentation (optimal value of $K$ and corresponding $K$-partition of the market) as interesting future work. We assumed a monopoly market. It will be interesting to study optimal fair pricing in the face of competition and other constraints such as finite supply, non-linear production cost, and variable demand.

### Acknowledgements

We thank the reviewers for valuable discussions on important topics related to this work and for helping us to improve it further.

This research is partially funded by the Department of Science & Technology, India, under grant number SRG/2020/001138.

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
