# OpenReview forum: " Individual Fairness in Feature-Based Pricing for Monopoly Markets "
_auai.org/UAI/2022/Conference — UAI 2022 Poster_

### Official Review · Reviewer_qDnt · 2022-04-08

**Q2(1) Originality/Novelty:** 1
**Q2(2) Significance/Impact:** 2
**Q2(3) Correctness/Technical Quality:** 3
**Q2(6) Clarity Of Writing:** 3
**Q6 Overall Score:** 5
**Q8 Confidence In Your Score:** 3

**Q1 Summary And Contributions:**

The paper introduces the individual fairness notion into the personalized pricing setting. They theoretically analyze the cost of the fairness constraints on the expected revenue under several mild assumptions. They further propose an algorithm to achieve an approximate best revenue under the continuous and concave revenue function assumption.

**Q2 Assessment Of The Paper:**

More detailed information regarding each of these aspects is given below:

**Q2(5) Reproducibility:**

3: Good: Key resources (e.g., proofs, code, data) are available and key details (e.g., proofs, experimental setup) are sufficiently well-described for competent researchers to confidently reproduce the main results.

**Q3 Main Strengths:**

1. The paper studies the fair personalized pricing problem, which is of significant importance nowadays.
2. The paper provides a comprehensive analysis of the cost of fairness under the $\alpha$-fairness constraint.

**Q4 Main Weakness:**

The main weakness of the paper is the lack of discussion with [1], which makes the contribution of the paper marginal.

1. The fairness constraint, though claimed as individual fairness by the authors, is actually defined on different segments of the population. As a result, it is a kind of group fairness and is the same as the price fairness proposed by [1].
2. [1] also theoretically analyze the effect of the fairness constraints on both revenue, consumer surplus, and total surplus. As a result, the authors should provide a detailed comparison with their theoretical results.

[1] Cohen, Maxime C., Adam N. Elmachtoub, and Xiao Lei. "Price Discrimination with Fairness Constraints." Proceedings of the 2021 ACM Conference on Fairness, Accountability, and Transparency. 2021.

**Q5 Detailed Comments To The Authors:**

In summary, I expect the authors to provide a detailed comparison with [1], including the differences in both definitions and theoretical results. There are also some other minor points
1. It would be better if the authors could provide more explanations on the concave assumption on the revenue. Specifically, what valuation distributions satisfy the assumption and, are these distributions common in practice?
2. It would be better if the authors could provide some experimental results to validate their theories.

[1] Cohen, Maxime C., Adam N. Elmachtoub, and Xiao Lei. "Price Discrimination with Fairness Constraints." Proceedings of the 2021 ACM Conference on Fairness, Accountability, and Transparency. 2021.


**Q7 Justification For Your Score:**

Although the theoretical analysis in the paper is convincing, I doubt the significance and novelty of the paper due to the lack of discussions with Cohen et al. As a result, I give a borderline reject in this round.

**Q9 Complying With Reviewing Instructions:**

1: Yes.

---

### Official Review · Reviewer_SBts · 2022-04-14

**Q2(1) Originality/Novelty:** 3
**Q2(2) Significance/Impact:** 3
**Q2(3) Correctness/Technical Quality:** 3
**Q2(6) Clarity Of Writing:** 3
**Q6 Overall Score:** 6
**Q8 Confidence In Your Score:** 2

**Q1 Summary And Contributions:**

This paper investigates individual fairness for price discrimination in a monopoly market. It defines a notion of individual fairness, then studies two settings, discrete valuation and continuous valuation. Finally, it provides feasible and optimal solutions for both settings

**Q2 Assessment Of The Paper:**

More detailed information regarding each of these aspects is given below:

**Q2(4) Quality Of Experiments (Optional):**

3: Good: The experimental evaluation is adequate, and the results convincingly support the main claims.

**Q2(5) Reproducibility:**

3: Good: Key resources (e.g., proofs, code, data) are available and key details (e.g., proofs, experimental setup) are sufficiently well-described for competent researchers to confidently reproduce the main results.

**Q3 Main Strengths:**

1. A new notion for fairness in pricing is defined. This notion is simple but coherent with the following solutions.
2. In the discrete valuation setting, the optimization is formulated as an integer programming task with an efficient solution.
3. The continuous setting has a solution with nice properties.

**Q4 Main Weakness:**

1. I believe experimental results would make the results more convincing.

I don't have major concerns about this article.

**Q5 Detailed Comments To The Authors:**

The notion of individual fairness is very similar to the notion proposed in R. S. Zemel, Y. Wu, K. Swersky, T. Pitassi, and C. Dwork, “Learning fair representations,” in ICML 2013. A reference is needed

**Q7 Justification For Your Score:**

I don't have expertise in pricing discrimination. Overall it looks promising to me and I don't see major flaws.

**Q9 Complying With Reviewing Instructions:**

1: Yes.

---

### Official Review · Reviewer_mjaE · 2022-04-18

**Q2(1) Originality/Novelty:** 2
**Q2(2) Significance/Impact:** 2
**Q2(3) Correctness/Technical Quality:** 2
**Q2(6) Clarity Of Writing:** 2
**Q6 Overall Score:** 5
**Q8 Confidence In Your Score:** 2

**Q1 Summary And Contributions:**

The paper considers individual fairness in the market pricing problem. The paper proposes a fairness notion called "\alpha-fairness", and shows theoretical results on the Cost of Fairness (CoF) ratio. A pricing algorithm is also provided.

**Q2 Assessment Of The Paper:**

More detailed information regarding each of these aspects is given below:

**Q2(5) Reproducibility:**

2: Fair: Key resources (e.g., proofs, code, data) are unavailable but key details (e.g., proof sketches, experimental setup) are sufficiently well-described for an expert to confidently reproduce the main results.

**Q3 Main Strengths:**

The merit of the paper lies in the effort of deriving bounds on CoF ratio (and the corresponding tightness analysis), as well as an algorithm to derive \alpha-fair pricing policy for the monopoly market.

**Q4 Main Weakness:**

There are several concerns regarding the presented results (detailed later).

1. The proposed fairness notion "\alpha-fairness" seems to be a rehash of the (metric-based) Individual Fairness notion proposed by Dwork et al.

2. The paper can benefit from a clearer presentation, distinguishing the novel contribution from the related literature.

**Q5 Detailed Comments To The Authors:**

Question 1: the connection between \alpha-fairness and Individual Fairness by Dwork et al.

To the best of my understanding, in the proposed notion of \alpha-fairness, we can directly view \alpha as a Lipschitz constant, and the proposed notion is identical to the Individual Fairness notion by Dwork et al. A further clarification regarding whether \alpha-fairness is only a rehash of previously proposed fairness notion is very necessary.

Question 2: the motivation of studying fairness in the monopoly market pricing scenario

While I understand the pricing scenario, it seems to me that the problem setup is exactly the same as the Individual Fairness audit w.r.t. a regression-based prediction/decision: the monopoly market pricing policy is the regressor (decision-maker), the price is the regression estimation, and the values of customers can be captured by distance metrics (might need to consider the features of individuals, etc.). I am not sure how the proposed approach contributes to the literature. How does the result differ from imposing Individual Fairness to this particular practical setting of market pricing?

----

Post-rebuttal update included.



**Q7 Justification For Your Score:**

The approach/notion presented in the paper seems to be a rehash of a previous result. Further clarification of the relation between this paper and previous literature would be very necessary. The overall score reflects the concern regarding the unclear presentation and limited contribution/impact.

---

Post-rebuttal:

Overall, there is no major flaw. Please kindly consider incorporating clarifications regarding the connection between Dwork et al. and the current work.

**Q9 Complying With Reviewing Instructions:**

1: Yes.

---

### Decision · Program_Chairs · 2022-05-15

**Decision:**

Accept (Poster)

**Comment:**

Meta Review: The paper proposes a new notion of individual fairness in personalized pricing setting. Generally the reviewers consider the theoretical contribution of this work sound and interesting, and the authors have provided several clarifications of the relation between this work and some prior literature. The authors are encouraged to incorporate the review comments for further improving the paper.